

# An improved global zenith tropospheric delay model GZTD2 considering diurnal variations

YiBin Yao[1,2,3], YuFeng Hu[1], Chen Yu[1], Bao Zhang[1], JianJian Guo[1]

[1]*School of Geodesy and Geomatics, Wuhan University,* 129 *Luoyu Road, Wuhan* 430079*, China. E-mail:* ybyao@whu.edu.cn

[2]*Key Laboratory of Geospace Environment and Geodesy, Ministry of Education, Wuhan University,* 129 *Luoyu Road, Wuhan* 430079*, China*

[3]*Collaborative Innovation Center for Geospatial Technology,* 129 *Luoyu Road, Wuhan* 430079*, China*

***Abstract--***The zenith tropospheric delay (ZTD) is an important atmospheric parameter in the wide application of GNSS technology in geoscience. Given that the temporal resolution of the current Global Zenith Tropospheric Delay model (GZTD) is only 24 h, an improved model GZTD2 has been developed by taking the diurnal variations into consideration and modifying the model expansion function. The data set used to establish this model is the global ZTD grid data provided by Global Geodetic Observing System (GGOS) Atmosphere spanning from 2002 to 2009. We validated the proposed model with respect to ZTD grid data from GGOS Atmosphere, which was not involved in modeling, as well as International GNSS Service (IGS) tropospheric product. The obtained results of ZTD grid data show that the global average Bias and RMS for GZTD2 model are 0.2 cm and 3.8 cm respectively. The global average Bias is comparable to that of GZTD model, but the global average RMS is improved by 3 mm. The Bias and RMS are far better than EGNOS model and the UNB series models. The testing results from global IGS tropospheric product show the Bias and RMS (-0.3 cm and 3.9 cm) of GZTD2 model are superior to that of GZTD (-0.3 cm and 4.2 cm), suggesting higher accuracy and reliability compared to the EGNOS model, as well as the UNB series models.

***Key Words—***Zenith tropospheric delay; GGOS Atmosphere; IGS; Diurnal variation; GZTD2 model.



## 1. Introduction


Radio space-based geodesy techniques suffer from atmosphere propagation delays,
of which the ionospheric delay can be largely eliminated by iono-free carrier phase
combination techniques, and then the tropospheric delay becomes the main error source.
In general, we project the slant delay to zenith direction with mapping function in GNSS
navigation and positioning, so modeling the ZTD is a common method to reduce the
tropospheric influence on signal travelling. In order to better exploit the modern
development of geodetic techniques, a more reliable tropospheric delay model is
required to improve the accuracy and efficiency of the application in earth science based
on space geodesy techniques.
The correction accuracy of some traditional tropospheric delay models such as
Hopfield model (Hopfield 1969), Saastamoinen model (Saastamoinen 1973), Black
model (Black 1978), can be up to centimeter or decimeter level using the real-time
meteorological parameters, while these models perform poorly when using the standard
atmospheric meteorological parameters. Collins and Langley (1997) established UNB
series models for the promotion of U.S. Wide Area Augmentation Navigation System
(WAAS). In North America, the average tropospheric zenith delay error of UNB3
model was 2 cm (Collins et al. 1998). UNB3m model estimates the wet delay using
relative humidity, and the average deviation was -0.5 cm (Leandro et al. 2006; Leandro
et al. 2008). EGNOS model is a tropospheric delay correction model used by European
Geostationary Navigation Overlay System (EGNOS), which is established by using the
$1°\times 1°$ grid data generated by the European Centre for Medium-Range Weather
Forecasts (ECMWF) (Dodson et al. 1999; Penna et al. 2001; Ueno et al. 2001), whose
correction accuracy is close to that of Hopfield and Saastamoinen model provided with
meteorological measurements. Li Wei et al. (2012) established the IGGtrop global
tropospheric delay empirical model using the three-dimensional parameter table from
reanalysis data of National Centers for Enviromental Predication (NCEP), which
considered the longitudinal changes of zenith troposphere. The accuracy was improved
significantly, but the calculation of zenith tropospheric total delay required a number of

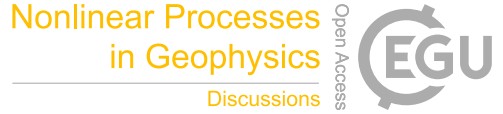

parameters. Then Li Wei et al. (2015) developed the new versions of IGGtrop named
IGGtrop_ri (i = 1, 2, 3) by simplifying the algorithm and lowering the resolution, which
substantially reduce the required numbers with a similar accuracy. Krueger (2004;2005)
and Schüler (2014) obtained the annual and diurnal coefficients for underlying
parameters by fitting every grid point's meteorological parameters time series of NCEP
atmospheric data, and established two global tropospheric delay models — TropGrid
and TropGrid2 with resolution of $1° \times 1°$. The correction accuracy of TropGrid2 is
slightly better than that of IGGtrop model. Böhm et al. (2015) proposed Global pressure
and temperature 2 wet (GPT2w) as an extension to GPT2 (Lagler et al. 2013) with an
improved capability to determine zenith wet delays in blind model. The GPT2w model
account for the annual and semiannual variations of meteorological parameters, and the
validation with IGS data and an extended validation with ray-traced delays (Möller et
al. 2014) show a high accuracy of about 3.6 cm for GPT2w. However, GPT2w has
numerous parameters for storage like above grid models such as IGGTrop series models
and TropGrid series models.
Yao et al. (2013) established a global non-meteorological parameters tropospheric
delay model GZTD (Global Zenith Tropospheric Delay) based on spherical harmonics
using the global zenith tropospheric delay grid data provided by Global Geodetic
Observing System (GGOS) Atmosphere. The harmonic function including three terms
(mean, annual and semi-annual) is used to fit the ZTD time series from 2002 to 2009
for each grid, then the fitted coefficients of all the girds are expanded with a 10-order
and 10-degree spherical harmonics. Its modeling approach was very simple, and the
overall accuracy of 4.2 cm was similar to the IGGtrop on a global scale, but the required
parameters were reduced greatly to about 600. GZTD model is constructed by global
daily average ZTD grid data and the model parameters were expanded with a low order
spherical harmonics, whose temporal resolution is only one day in theory and spatial
resolution is low.
In this paper, using the ZTD grid data provided by the GGOS Atmosphere, the
diurnal variations in ZTD were analyzed to prove the theoretically necessity for
temporal resolution improvement of GZTD model. Then on the basis of GZTD model



and taking the diurnal variations into consideration and modifying the expansion
function, we developed an improved global non-meteorological parameters ZTD model
— GZTD2. The data set used to establish this model is the global ZTD grid data
provided by the GGOS Atmosphere from 2002 to 2009. Using ZTD grid data obtained
from GGOS Atmosphere and tropospheric product (Buyn et al. 2009) provided by IGS
for model validation, the accuracy of GZTD2 model is superior to that of GZTD model,
and this model performs much better than other commonly used models such as
EGNOS model and UNB series models.

## 2 The new tropospheric delay model

The GGOS Atmosphere is a project that aims to establish atmospheric models,
which has been carried out at Vienna University of Technology and has been funded by
the Austrian Science Fund (Böhm & Schuh 2013). It provides grid data of global zenith
delays (including zenith hydrostatic delay (ZHD) and zenith wet delay (ZWD)) with
temporal resolution of 6 hours (0:00, 6:00, 12:00, 18:00UTC) and spatial resolution of
$2.5° \times 2°$ (lon×lat), which are derived from the reanalysis data (Uppala et al. 2005)
provided by the ECMWF. The ZTD grid data can be obtained by simply adding up the
ZHD and the ZWD at the same point and time. In this paper, the research about model
establishment is based on the ZTD grid data.

### 2.1 Diurnal variations in ZTD

Yao et al. (2013) developed a new global zenith tropospheric delay model (GZTD),
which is based on spherical harmonics without using meteorological parameters. GZTD
model depends on four parameters: the day of year (doy), the latitude, the longitude and
the height; and the overall accuracy is up to centimeter level. However, the algorithm
of GZTD model only considers the annual and semiannual cycles in ZTD and the
establishment of GZTD model is based on the daily average of global grid ZTD data,
hence the temporal resolution of GZTD model is one day (24 h) in theory. We randomly





selected six grid points which represent the regions in low, middle and high latitude in
both the southern and northern hemispheres respectively, and applied GZTD model to
estimate the ZTD at four moments (0:00,6:00,12:00,18:00 UTC) of the first doy in 2010,
then compared the GZTD model estimations with the corresponding data from GGOS.
The results are shown in Figure 1.

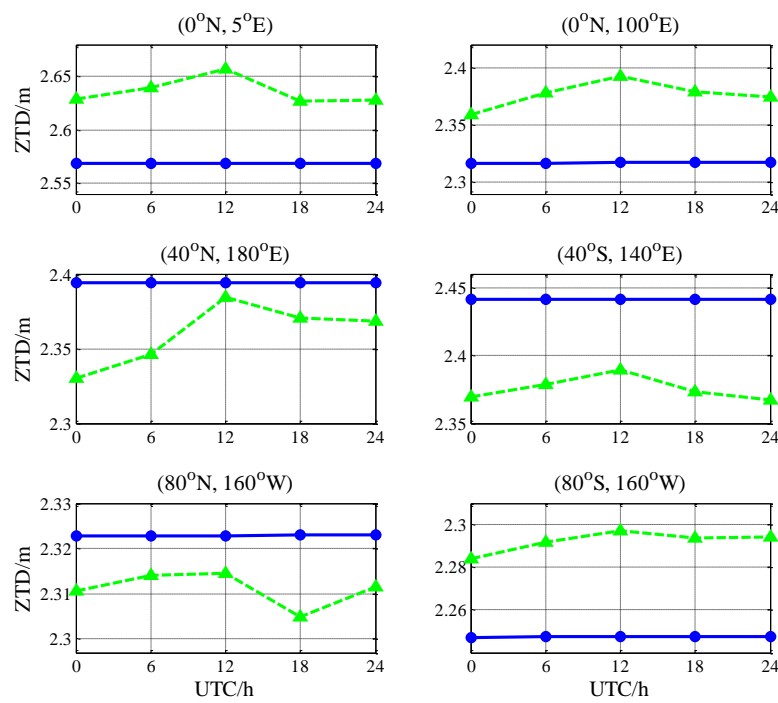


**Figure 1.** GZTD model estimates(blue ○ ) and corresponding GGOS grid values (green △ ) at the

first doy of 2010

We can see clearly from Figure 1 that the ZTD estimates of GZTD model can

almost be fitted with a straight line parallel to the time axis which only varies about 1
mm in a single day. The real variations of GGOS grid ZTD data are mostly up to
centimeter level, which is one order larger than the variations of GZTD model estimates.
Furthermore, we calculated the mean diurnal ZTD values of these six GGOS grid points
over the whole 2010 year (Figure 2), and the significant signal of diurnal variation can
be seen at all these six grid points .We can draw a conclusion that GZTD model could
not reflect the characteristic of diurnal variations in ZTD, so the model estimations
nearly have no difference when doing calculation with real value or corresponding



integer value of the input doy. Therefore, it is necessary to improve the temporal
resolution of GZTD model to reflect diurnal variations. It should be noted that Jin et al.
(2009) has investigated the diurnal and semidiurnal variations in ZTD which obtained
from a decade of global GPS observations, and thought that the atmospheric tides were
the major driver of these variations after finding the general similarities of diurnal
variations between ZTD and pressure. However, the semidiurnal variations could
hardly be described because of the low temporal resolution (6 h) of GGOS ZTD data,
so we didn't consider the semidiurnal components of ZTD in modeling in the following
section.

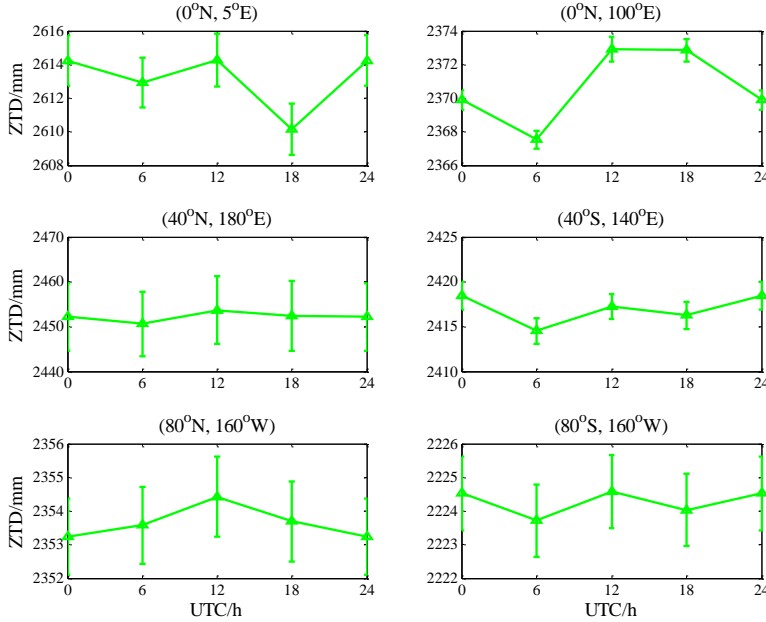


**Figure 2.** Mean diurnal ZTD values of GGOS grid points with error bars over the 2010 year

## 2.2 Establishment of GZTD2 model

According to the previous researches conducted by Jin et al. (2007) and Yao et al.
(2013), ZTD decreases exponentially with increasing height, and is featured by one-
year periodicity and half-year periodicity, and has a strong correlation with latitude.



Based on these characteristics of ZTD, we took diurnal periodic variations into
consideration to develop an improved model GZTD2. The expression of GZTD2 model
is as follows:
$$\mathrm{ZTD} = \left[ a_0 + a_1 \cos(2\pi \frac{\mathrm{doy}-a_2}{365.25}) + a_3 \cos(4\pi \frac{\mathrm{doy}-a_4}{365.25}) + a_5 \cos(2\pi \frac{\mathrm{hod}-a_6}{24}) \right] \exp(\beta h)$$

(1)

Where,
$$a_i = \sum_{n=0}^{18} \sum_{m=0}^{n} \mathrm{P}_{nm}(\sin\varphi) \cdot [\mathrm{A}^i_{nm} \cos(m\lambda) + \mathrm{B}^i_{nm} \sin(m\lambda)] \qquad (i=0,1,\cdots,6) \qquad (2)$$

In equation(1), doy is the day of year; hod is the UTC time; $h$ is the height
(altitude); $a_0$ is the annual mean of ZTD on the mean sea level (MSL); $a_1$ is the annual
variation amplitude of ZTD; $a_2$ is the initial phase of annual variation; $a_3$ is the
semiannual variation amplitude of ZTD; $a_4$ is the initial phase of semiannual variation;
$a_5$ is the diurnal periodic variation amplitude of ZTD; $a_6$ is the initial phase of diurnal
variation; $\beta = -0.00013137$ is the constant to reduce the ZTD at height to the MSL,
which was determined by Yao et al. (2013) by fitting the global GGOS grid ZTD via
exponential function with respect to height; $\mathrm{P}_{nm}$ is the Legendre polynomials; $\varphi$ is the
latitude of grid point; $\lambda$ is the longitude of grid point; $\mathrm{A}^i_{nm}$ and $\mathrm{B}^i_{nm}$ are the
coefficients of spherical harmonics determined by least square optimization.
For each grid-point-specific ZTD time series derived from GGOS Atmosphere, we
used equation (1) to fit them to temporal coefficients at MSL. However, there are seven
coefficients for each grid, which need large storage space on global scale. Then
referring to the idea of spherical harmonics used in GPT (Böhm et al., 2007), we used
equation (2) to express the temporal coefficients (mean, annual terms et al) of all grids
as a function of location (latitude, longitude and height), thus reducing the parameters.
Different from the GZTD model established using daily average global ZTD data, we
utilized the ZTD time series data of four moments per day (0:00, 6:00, 12:00, 18:00UTC)
from 2002 to 2009, provided by GGOS Atmosphere, to fit ZTD values to obtain
temporal variation parameters via equation (1), then expanded these parameters with a
18-order and 18-degree spherical harmonic function (equation (2)), respectively. We
used this spherical harmonic function instead of the 10-order and 10-degree function
adopted in GZTD model because it is not sufficient to apply the previous 10 order
function for the expansion of the temporal variation parameters with relatively high
resolution. The number of order and degree of spherical harmonics determine the
horizontal resolution of model. However, higher order and degree bring more
parameters for model. The resolution of GZTD model is about 18 ° while the diurnal
variations are mostly less than 5 mm. The 10 spherical harmonics are too low for
GZTD2 model to reflect the diurnal variations. To keep a balance between the
resolution and number of parameters, we used 18 spherical harmonics for GZTD2
whose resolution is about 10 °.

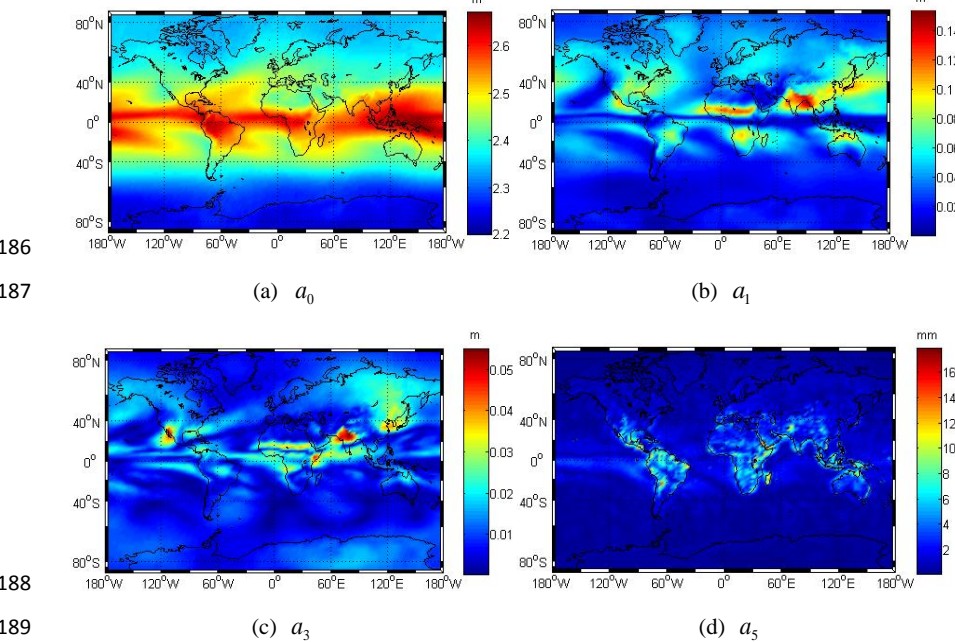


(a)   $a_0$           (b)   $a_1$


(c)   $a_3$           (d)   $a_5$

**Figure 3.** The global distribution of the annual mean ZTD on MSL (a) , the annual variation
amplitude (b), the semiannual variation amplitude (c), and the diurnal variation amplitude (d)
Figure 3 shows the global distributions of the annual mean of ZTD on MSL and



amplitude parameters after fitting by equation (1). As can be seen from Figure 3a, the
coefficient $a_0$ in low latitudes, especially, near the equator, are significantly larger
than that in high latitudes, and the distribution in the Southern Hemisphere is more
uniform than that in the Northern Hemisphere; These results are mostly in agreement
with the results of Li et al. (2012) and Yao et al. (2013). For the sawtooth shape in the
40 °N-40 °S region, Yao et al. (2013) found this shape appear in coastal areas and is
consistent with the directions of equatorial trade winds, so they assumed that the
distributions of ZTD are effected by some physical impacts such as terrains and heat
circulation. Compared with the previous discovery, the sawtooth shape in Figure 3a is
more evident, indicating that GZTD2 model incorporates these physical impacts.
Figures 3b and 3c show the global distributions of annual amplitude and semiannual
amplitude respectively, both of which are more uniform in the Southern Hemisphere
than that in the Northern Hemisphere, which is probably due to the fact that most parts
of the Southern Hemisphere are covered by oceans, while the Northern Hemisphere has
many seacoast regions which lead to relatively complex spatial variation.
Figure 3d shows the global distribution of diurnal variation amplitudes. It can be
seen that diurnal variation amplitudes are less than 3 mm in most parts of the world, but
up to centimeter in some low-latitude equatorial areas such as Central America, South
America, central Africa and tropical Asia, indicating notable diurnal variations in these
areas. The distribution characteristics of diurnal variation amplitudes is similar to the
results of Jin et al. (2009). So taking these diurnal variations into consideration in
GZTD2 model is quite reasonable and necessary in theory.
GZTD2 model only needs doy, UTC time, latitude, longitude and height as input
parameters in practical application. GZTD2 uses equation (2) to derive temporal
parameters $a_0$, $a_1$, $a_2$, $a_3$, $a_4$, $a_5$, $a_6$, which are then entered into equation (1)
together with the doy to get the ZTD at MSL. The realization of GZTD2 model is simple
with a few parameters, and the calculation is convenient without inputting any real-time
meteorological parameters. Table 1 summarizes the main improvements and features of
the newly suggested model compared to the GZTD model.



**Table 1.** Improvements of GZTD2 with respect to GZTD

|  | GZTD | GZTD2 |
|---|---|---|
| Data | Daily average ZTD grid data from GGOS: 2002~2009 | ZTD grid data with a resolution of 6 h from GGOS: 2002~2009 |
| Representation | Spherical harmonics up to degree 10 and order 10 | Spherical harmonics up to degree 18 and order 18 |
| Temporal variability | Mean, annual, and semi-annual terms | Mean, annual, semi-annual and diurnal terms |
| Horizontal resolution | About 18° | About 10° |


## 3. Validation and Analysis of GZTD2 model

To analyze the effectiveness and reliability of the new model and verify its
accuracy and stability on global scale, as well as to compare it with the GZTD model,
this section will exploit some data sources to conduct model validation. Two kinds of
data sources are used here, the first is ZTD grid data from GGOS Atmosphere which is
not used in modeling. The other is tropospheric product data provided by IGS. The
accuracy is characterized with the average deviation (Bias) and root mean square (RMS)
which are usually used for model validation (Yao et al., 2013; Li Wei et al., 2015; Böhm
et al., 2015). The expressions of Bias and RMS are:
$$Bias = \frac{1}{n}\sum_{i=1}^{n}(ZTD_i^M - ZTD_i^0) \qquad (3)$$

$$RMS = \sqrt{\frac{1}{n}\sum_{i=1}^{n}(ZTD_i^M - ZTD_i^0)^2} \qquad (4)$$

Where $ZTD_i^M$ is the value estimated by model and $ZTD_i^0$ is the reference value.

## 3.1 Validation with GGOS Atmosphere ZTD grid data

Data provided by GGOS Atmosphere from 2002 to 2009 are involved in modeling,
so we used the data of 2010 to test it. Since the resolution of ZTD grid data is 2°×2.5°,





the total number of grid points is 13,104. Treating the ZTD data at 0:00, 6:00, 12:00
and 18:00 UTC of everyday on each grid point as the reference values, we calculated
the bias and RMS of GZTD2, GZTD, EGNOS, UNB3 and UNB3m models. Statistical
analyses are shown in Table 2.
**Table 2.** Modeling errors of different models validated by GGOS data

|  | Bias (in cm) | | | RMS (in cm) | | |
|---|---|---|---|---|---|---|
|  | Mean | Min | Max | Mean | Min | Max |
| GZTD2 | 0.2 | -3.7 | 6.2 | 3.8 | 0.9 | 8.3 |
| GZTD | 0.2 | -5.4 | 8.0 | 4.1 | 1.1 | 9.5 |
| UNB3m | 3.3 | -7.2 | 16.0 | 6.4 | 1.3 | 16.5 |
| UNB3 | 4.5 | -7.0 | 16.7 | 7.0 | 1.1 | 16.9 |
| EGNOS | 4.5 | -9.6 | 17.7 | 7.2 | 1.0 | 18.1 |

As can be seen from Table 2, for the total 13104 points involved in the global
validation, GZTD2 model's mean Bias is 0.2 cm with a maximum of 6.2 cm, and the
average of RMS is 3.8 cm with a maximum of 8.3 cm, significantly better than the
EGNOS and UNB series models, and the RMS is reduced by 3 mm compared with that
of GZTD model. UNB3m model's accuracy is about 1 cm better than UNB3 and
EGNOS models, so we only chose UNB3m as the representative of commonly used
model in our following comparison analysis. Figure 4 shows the global distributions of
Bias and RMS of the three models.
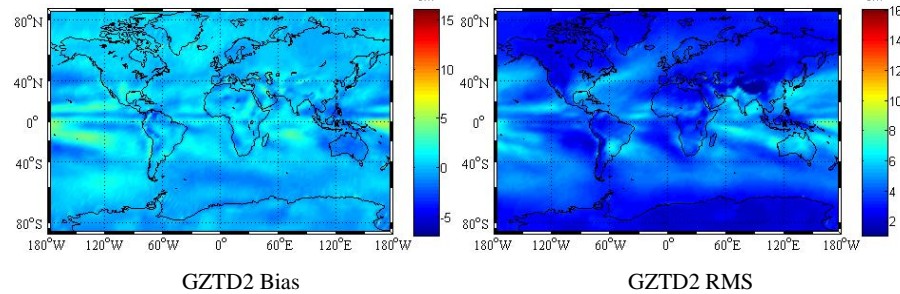

GZTD2 Bias                           GZTD2 RMS





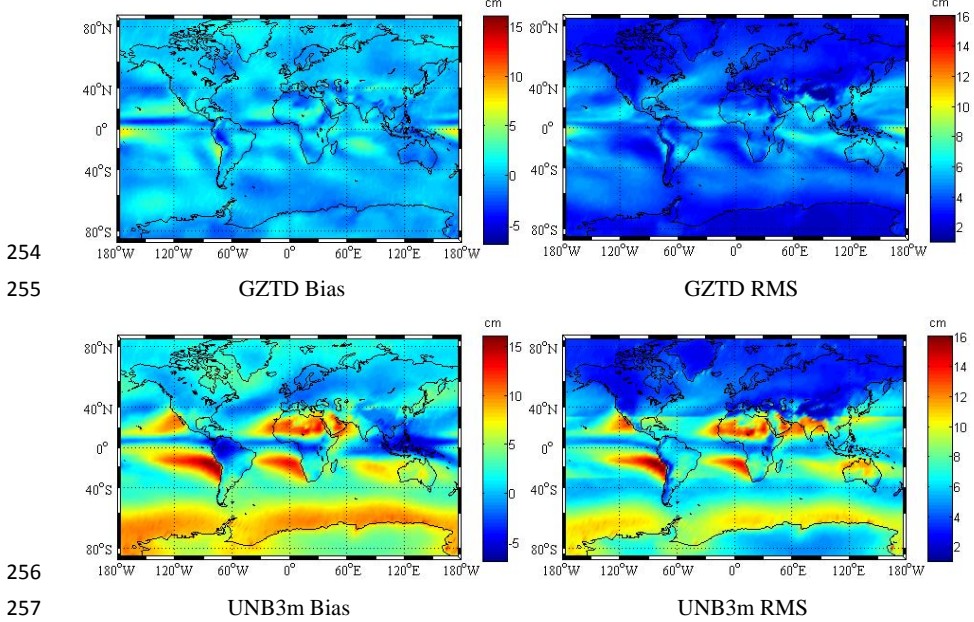

**Figure 4.** Global distribution of Bias and RMS of different models

As can be seen from Figure 4, compared with the other two models, the new model has better accuracy in the world wide scale, and the accuracy of the areas where lager errors appear improves significantly. Compared with GZTD model, GZTD2 model improves the accuracy in the equator area. Obviously, all these three models have suffered large errors in the Pacific Ocean near the equator and Indian Ocean. These areas are near the equator and may be affected by trade winds and ocean currents, so the climate change in these areas are more complex compared with other areas, resulting in difficulty for modelling tropospheric delay. In addition, GZTD2 and GZTD model are comparable in Northern and Southern Hemispheres, but the UNB3m model's accuracy is obviously lower in the Southern Hemisphere, this is because the UNB3m model is based on the assumptions that tropospheric delay is symmetrical with equator (Leandro et al., 2006). In fact, this assumption is not reasonable enough and the modeling data source are derived from North America, so the accuracy of the model is higher in North Hemisphere, especially in Northern America.



## 3.2 Validation with IGS tropospheric delay data

IGS has provided final troposphere products with a temporal resolution of 5 minutes since 1998. There are 362 IGS sites selected in 2010 to verify the accuracy of GZTD2 model, and the distribution of IGS sites is shown in Figure 5. The uncertainties of the ZTD products are very small (see Figure 5) with a mean value of 1.5 mm, indicating high quality of the ZTD products. Considering the ZTD products of IGS sites as true value, we tested and analyzed the ZTD estimates of GZTD2 model, GZTD model, EGNOS model and UNB series models. The Bias and RMS statistical results are shown in Table 3.

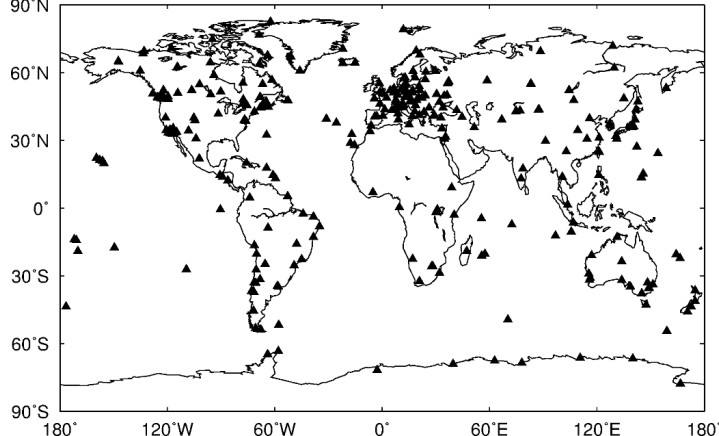

**Figure 5.** Distribution of global IGS sites involved in validation

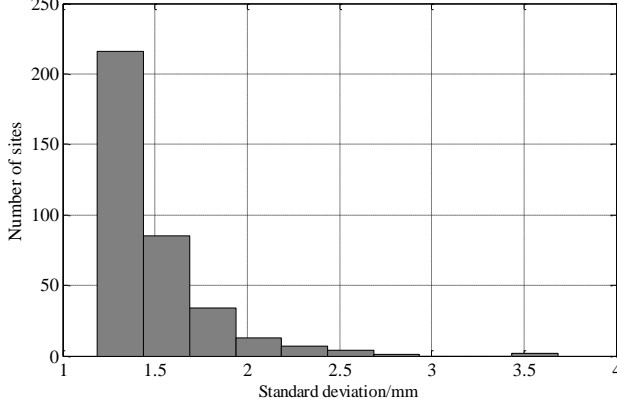

**Figure 6.** Histogram of uncertainty of ZTD at selected IGS sites






**Table 3.** Error of different considered models versus IGS data

|  | Bias (in cm) | | | RMS (in cm) | | |
|---|---|---|---|---|---|---|
|  | Mean | Min | Max | Mean | Min | Max |
| GZTD2 | -0.3 | -5.4 | 3.2 | 3.9 | 2.0 | 8.3 |
| GZTD | -0.3 | -6.0 | 5.1 | 4.2 | 2.1 | 8.5 |
| UNB3m | 1.2 | -6.7 | 11.2 | 5.2 | 2.4 | 12.2 |
| UNB3 | 2.6 | -6.5 | 13.4 | 5.6 | 2.3 | 13.7 |
| EGNOS | 2.4 | -6.6 | 15.3 | 5.7 | 2.4 | 12.3 |

As can be seen from Table 3, in terms of the results of accuracy and stability testing
for all IGS sites throughout the year, GZTD2 model performs with the best average
RMS, and then GZTD model followed. Global correction accuracy of the new model
reaches centimeter level: Bias average value is -0.3 cm, average RMS is 3.9 cm.
Compared with GZTD model, the range of Bias of GZTD2 model reduce by 2.4 cm
and the maximum RMS of GZTD2 model decreases by 0.2 cm, indicating that the new
model has a higher stability. Bias and RMS of EGNOS model are very close to those
of UNB3 model and both are worse than UNB3m, which is similar to the results of Li
et al. (2012). To display the correction effects of different models in a more intuitive
way, we computed the distributions of Bias and RMS of all IGS stations. Figure 7 shows
the histograms of Bias and RMS for the three models.





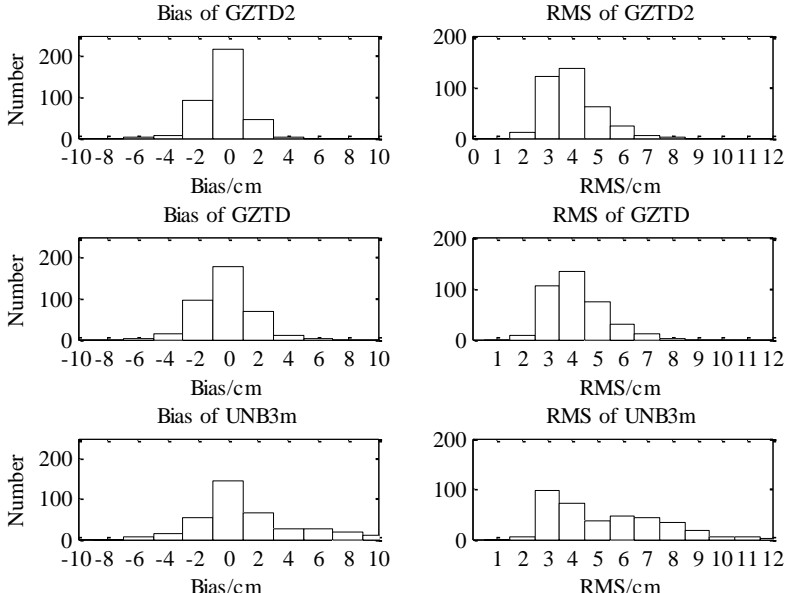


**Figure 7.** Histograms of Bias and RMS for three models
As can be seen from Figure 7, the Bias of GZTD2 model concentrates in range of
[-3cm 3cm], while the main distribution range of the Bias of GZTD model are 1cm
larger, and the Bias for UNB3m is distributed with the range more than 8 cm. It indicates
that GZTD2 model and GZTD model have small systematic deviations compared with
IGS data on a global scale, with the former performing better than the latter, but
problematic systematic deviations exist in the UNB3m model within some special areas.
Figure 7 also shows that the RMS of GZTD2 model is mostly around 4 cm, whose
distribution is more concentrated compared to GZTD model, indicating GZTD2 model
has higher stability than GZTD. The RMS of UNB3m model are mainly around 5 cm
and exceed 9 cm at many sites, which further suggests the existence of systematic
deviations in certain areas in the UNB3m model.



(a) GZTD2 Bias                                        (b) GZTD2 RMS

(c) GZTD Bias                                        (d) GZTD RMS

(e) UNB3m Bias                                       (f) UNB3m RMS

**Figure 8.** Global distributions of Bias and RMS for different models

To further analyze the accuracy of the different models varying with location,

Figure 8 shows the global distributions of Bias and RMS calculated from different
models for IGS sites. As can be seen from Figure 8, GZTD2 and GZTD model largely
eliminate the effects caused by latitude and longitude variations, and the former is more
stable than the latter in terms of global distribution of Bias and RMS in spite of a few
sites with relative large error, of which most sites are located in the ocean and seacoast



areas. A more clear comparison in terms of RMS between GZTD and GZTD2 is shown
in Figure 9.The reduce for RMS can be found at most sites (the number is 273) when
moving from GZTD to GZTD2, which account for 75.4% of all sites. The significant
improvements of RMS are found at the sites in low-latitude areas such as Pacific Ocean,
South America coast and West Africa coast where the diurnal variations are notable
(see Figure 3d). This result proves the reasonability of adding diurnal variations in
GZTD2. For UNB3m model, as it is presented in Figure 8 Biases are negative in most
parts of the Northern Hemisphere and positive in most parts of the Southern
Hemisphere with significantly larger deviations, and RMS are smaller for areas in the
latitudes higher than 30 degrees, again suggesting that the correction effect of UNB3m
model is regional.

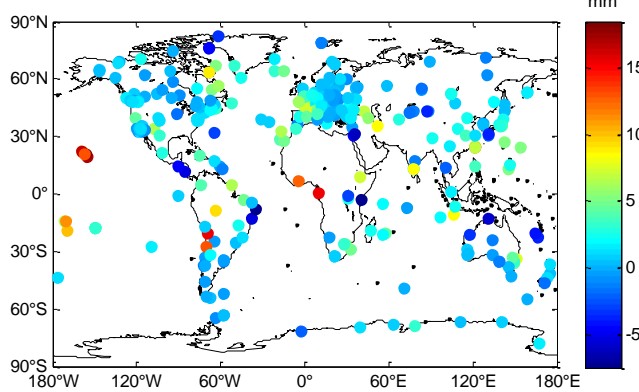


**Figure 9.** Global distribution of the difference between GZTD's RMS and GZTD2's RMS (GZTD's
RMS minus GZTD2's RMS)
Figure 10 shows the global distribution of Bias and RMS with respect to height
for GZTD2 model, GZTD model and UNB3m model. As can be seen, the Bias and
RMS are lager with height less than 500 m for all three models. Between 500m and
2000m height, the Bias and RMS of GZTD model and GZTD2 model perform better
than that of UNB3m model, and the overall correction effects of the GZTD and GZTD2
model are also better than the latter. Due to the same exponential function and reducing
constant for height, the distribution patterns of the Bias and RMS of GZTD and GZTD2
model with respect to height are roughly similar, but the latter is obviously superior to



the former.

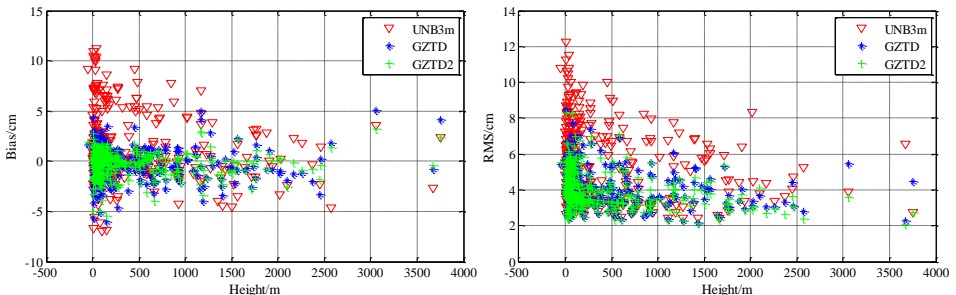

**Figure 10.** Global distributions of Bias and RMS for different models with respect to height
For a more comprehensive analysis of the relationship between model stability and
height, Figure 11 presents the global distribution of relative RMS for three models with
respect to height. The relative RMS is the ratio of the RMS to the annual mean ZTD at
the site. Basically, a relative accuracy between 1% and 2.5% can usually be stated for
the majority of the sites from GZTD2 model, and the relative accuracy is less than 3%
for GZTD model, showing that both perform better than UNB3m model.

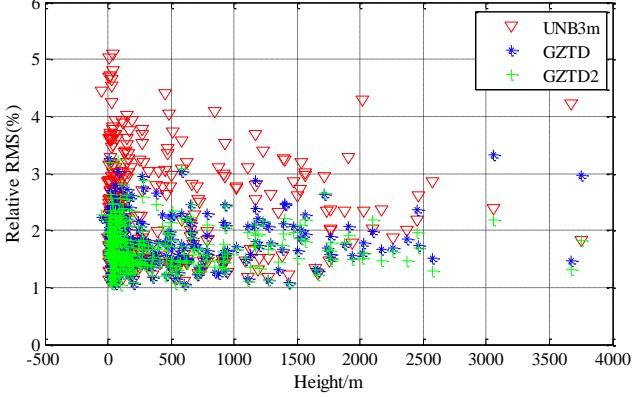

**Figure 11.** Relative RMS for different models with respect to height
Figure 12 illustrates the comparisons between IGS ZTD data and ZTDs
determined by UNB3m, GZTD and GZTD2 models over the year 2010 at site KOUR
and TWTF. During the whole year 2010, the ZTD values estimated by GZTD2 model
show the best agreement with the IGS data, which are better than that of GZTD model
without diurnal terms. The ZTDs determined by UNB3m model vary slightly
throughout the year 2010, thus resulting in poor performance. The results in Figure 12
indicates that GZTD2 model has a temporal stability for correction accuracy.

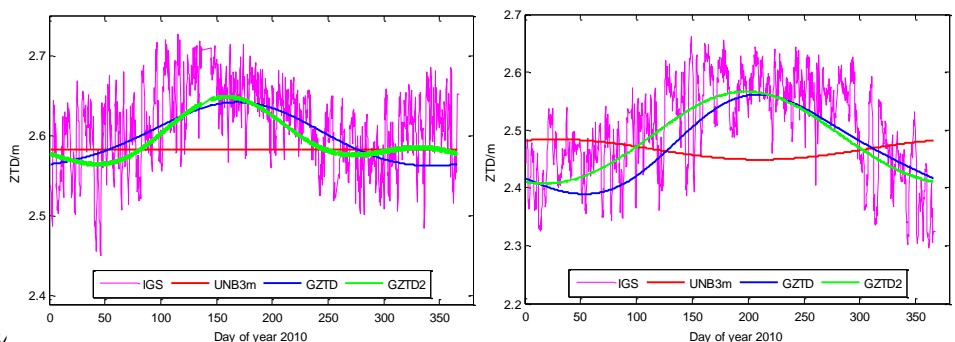


**Figure 12.** ZTDs at site KOUR (5.3 °N, 52.8 °W, 9.5m; left) and TWTF (24.9 °N, 121.2 °E, 189.9m;
right) as provided by IGS and as estimated by different models over year 2010

From the above analysis, we can conclude that the overall accuracy of GZTD2

model is up to centimeter level. GZTD2 model is obviously superior to other commonly
used models in terms of Bias and RMS, and the accuracy improve significantly
compared with GZTD model, thus performing a higher reliability and stability.

## 372    4 Conclusions

In this paper, we used time series data of global tropospheric zenith delays

provided by GGOS Atmosphere, and considered the diurnal variation in the ZTD based
on the GZTD model, and adopted a modified expansion function, and ultimately
developed an improved model named GZTD2. We conducted external validation
testing with ZTD grid data which was not involved in modeling, and IGS tropospheric
product. The testing results of ZTD grid data reflect the global precision and stability
for GZTD2 model at four moments each day, and the global average Bias and RMS for
GZTD2 model are 0.2 cm and 3.8 cm respectively; the global average Bias is
comparable to that of GZTD model, but the global average RMS has been reduced by
3 mm; the Bias and RMS are far better than EGNOS model and the UNB series models.
The testing results of global IGS tropospheric product show the Bias and RMS for
GZTD2 model are -0.3 cm and 3.9 cm, superior to that of GZTD (-0.3 cm and 4.2 cm),

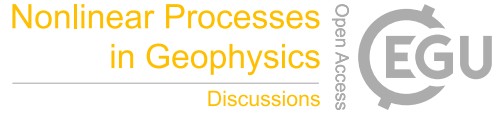

indicating higher accuracy and reliability compared to the EGNOS model and the UNB
series models.
Overall, compared to GZTD model, GZTD2 model improves the temporal
resolution and spatial resolution by considering diurnal periodic variations and
modifying the expansion function, further completing and optimizing the theory of
model establishment. The reliability and stability for GZTD2 model are much better
than other commonly used models. However, like other empirical models such as
UNB3m, GZTD2 model would be inaccurate in extreme weather events. Saastamoinen
model is recommended if the real-time meteorological observations are available under
extreme weather events. Moreover, GZTD2 model doesn't consider the semidiurnal
variations due to the temporal resolution of GGOS data. In order to build a global
tropospheric model with high accuracy, ZTD data with high quality and resolution are
required, and the diurnal and semidiurnal variations as well as the subtle secular
variation trend of ZTD need more detailed and further study.

*Acknowledgements The authors would like to express gratitude toward GGOS*
*Atmosphere for providing related data. Great appreciation also goes to IGS for their*
*data supporting of reference tropospheric product. This research was supported by the*
*National Natural Science Foundation of China (41174012; 41274022) and The*
*National High Technology Research and Development Program of China*
*(2013AA122502) and the Fundamental Research Funds for the Central Universities*
*(2014214020202) and the Surveying and Mapping Basic Research Program of*
*National Administration of Surveying, Mapping and Geoinformation (13-02-09).*

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
