# Peer review of "An improved global zenith tropospheric delay model GZTD2 considering diurnal variations"

_Nonlinear Processes in Geophysics, 2016_

## Referee Comment (RC1) · Anonymous Referee #1 · 25 Apr 2016

**Review: An improved global zenith tropospheric delay model GZTD2 considering diurnal variations**

The paper describes an improved method to compute the Zenith Tropospheric Delay, based on statistics of ZTD data on the global scale. It is an updated version of a previous model, which now includes the effects of the diurnal cycle.

**1  General comments**

From a general point of view, the addition made by this model are well presented and the results show that the accuracy of the model improves.

The text is comprehensible everywhere, however, the manuscript would benefit from copy-editing for English grammar and usage.

Subject to the comments below, the manuscript would be appropriate for publication.

**2  Scientific questions**

ln 32-33 – '...  by iono-free combination techniques': a citation about such techniques should be useful, given the general audience to which the journal refers to.

ln 37 – 'A more reliable'. The use of this expression would imply the fact that a model has already been presented above in the text (maybe 'more' could be dropped). Also, the use of 'In order to better exploit the modern development of geodetic techniques' and ' to improve the accuracy and efficiency of [...] based on geodesy technique' in the same phrase is a bit redundant.

line 40-57 – A (very) short description on the basic principles on which each model presented here is based (real time data, averaged trend from database, empyrical considerations...) would be helpful, along with their accuracy (which is already present).

ln 58 – '... required a number of' Which number? A large number of parameters? (If I correctly understand the context)

ln 87 – 'the theoretical necessity'. I think that the use of analysis of data should show more the practical necessity, or the expression can just be omitted.

ln 116 – Do they represent the regions in the sense that they are representative? Or are they just example on how an expected diurnal cycle at different latitude should qualitatively be?

ln 118 – The first time that day of the year (doy) is written, it should be written in the extended form.

Figure 2, captions and/or description – It is not very clear what error are represented by the vertical bars. Are they one standard deviation from the average?

lines 171 - 179 – The fact that the new model is presented before its previous version makes this part not too easy to follow, at first. Maybe a swap between the two sentences, where the features of the older model are presented before the new ones could simplify things.
If I may add a comment, since the previous model has appeared only in (Yao et al.,2013) which is accessible only to chinese readers, a short introduction of this model at the beginning of this section (like the counterpart of formulae 1 and 2) can be useful.

line 181 – I don't get what this sentece mean, exactly. It is stated that the GZTD horizontal resolution is 18 degrees, but it is not clear if the diurnal variation above 5mm refers to the GZTD model or the expected diurnal variation which the model should usually solve. There are a couple of questions that should be answered, even if shortly, to clarify this part: how the horizontal resolution and ZTD resolution are related and what should be an optimal horizontal accuracy to get the most of the diurnal effects?

ln 214 – I would drop the last 'in theory'.

ln 265 – I am confused by the term 'climate change'. Do you mean that the statistics of the previous years that you include to set the parameter of GZTD2 and the other model are not representative of the year that you are using for the test? To me, the fact that the change in the ZTD has a relation with deep moist convection effects, whose specific occurence is impossible to predict using averaged data, and that are much more intense at the tropics, would seem sufficient to explain the increased error near the equator, at least qualitatively (so that they are related more to weather changes than to climate changes). If you are connecting it to climate change please explain better, possibly including some references.

line 345 and line 368 – '... is obviously superior ...': I think that, even if 'obviously' is here clearly intended with the meaning of 'easily seen', its more common meaning of 'of course' can be a little misleading. I suggest to change it in some way.

ln 373 - 376 – This phrase needs some adjustment, with more clear logical connections between the part that are now simply separated by commas.

ln 378 – The sentence starting with 'The testing results ...' is not very clear. I suggest to simplify and organize the informations better.

**3 Techincal corrections**

line 12 – Even if somewhat well known, I think that GNSS should be expressed with its full name at least one time.

ln 20 – It should be written Root Mean Square the first time it is used.

ln 56 – Predication → Prediction

ln 72-73 – '... parameters for storage like *the* above grid models ...'

ln 79 – girds → grids

ln 146 – '..., and is featured ...': 'and' should be removed.

ln 162 – '$P_{nm}$ is ... ' → '$P_{nm}$ are ... '

ln 171 – 'Different from' → 'In contrast with'.

ln 215 – 'GZTD model ...' → 'The GZTD model ...'

ln 289 – 'followed' → 'follows'.

ln 325 – 'The reduce for RMS ...' → 'The reduction of RMS ...'

ln 340 – 'the Bias and RMS are lager with height less than 500 m ...' → 'the Bias and RMS are larger for height less than 500 m ...'

ln 369 – 'improve' → 'improves'

ln 377 – 'testing with ZTD grid data' → 'testing with GGOS ZTD grid data' (since in this statement both the sources of data used are explicitly listed).

ln 383 – 'show the Bias ...' → 'show that the Bias'

ln 384 – 'that of GZTD' → 'those of GZTD'

---

## Referee Comment (RC2) · Anonymous Referee #2 · 28 Apr 2016

Title: An improved global zenith tropospheric delay model GZTD2 considering diurnal variations Author(s): Y. Yao, Y. Hu, C. Yu, B. Zhang, and J. Guo MS No.: npg-2016-9 MS Type: Research article

The paper describes an improved method to compute the Zenith Tropospheric Delay including the effects of the diurnal cycle based on statistics of ZTD data on the global scale. The model is well presented and the results show that the accuracy of the model improves.

**1 General comment**

Nonlinear Processes in Geophysics (NPG) is an international, interdisciplinary journal for the publication of original research furthering knowledge on nonlinear processes in all branches of Earth, planetary, and solar system sciences. The article not discuss

about multifractals, turbulence, complex systems, nonlinear waves, pattern formation, complex networks, stochastic processes, extreme events, bifurcation, chaos, phase transitions, complex systems. Aspects related with scaling, predictability and data assimilation are indirectly commented.

As manuscript, the article would be appropriate for their publication.

2 Scientific questions

Reading becomes tedious by overuse of acronyms and references to other models are results. Please, use cm or mm, not both.

I think that it's necessary an atmospheric approximation to problem. An approximation as of the article 'Seasonal variability of GPS-derived zenith tropospheric delay (1994–2006) and climate implications' of Shuanggen Jin, Jong-Uk Park, Jung-Ho Cho, Pil-Ho Park in Climate and Dynamics Journal can be more appropriate for this journal. A reference to this article can be included.

An article very similar has been written for Li Wei with their IGGTrop model for Chinese Science Bulletin, and comparisons with results of IGGtrop model (or TopGrid2) can be of interest when you compare your results. Please, change in your references and include the entire name of first author (line 447, line 449).

The aspects related with validation (3.2), can be the most contentious. Why you use these 362 IGS sites?.

Aspects related with the ZTD changes are poorly traited. Convection effects are much more intense at the tropics daily. This would seem sufficient to explain the increased error near the equator. In line 327 you compare between GZTD and GZTD2 and you indicate that there are improvements of RMS in areas where you have limited IGS sites to compare.

3 Technical corrections

Change Predication in line 56 for Prediction. Change girds for grids in line 79. Include '2. The new..' in line 98 and '4. Conclusions' in line 372. You forgot the dot.

---

## Author Comment (AC1) · 29 Apr 2016

**Scientific questions**

1. ln 32-33 – '... by iono-free combination techniques': a citation about such techniques should be useful, given the general audience to which the journal refers to.

Response: Thanks for your comment. We have added a citation about iono-free combination techniques as you suggested.It is 'Spilker, J. J. (1980). GPS signal structure and performance characteristics.Global positioning system, 1, 29-54.'

2. ln 37 – 'A more reliable'. The use of this expression would imply the fact that a model has already been presented above in the text (maybe 'more' could be dropped). Also, the use of 'In order to better exploit the modern development of geodetic techniques' and' to improve the accuracy and efficiency of [...] based on geodesy technique' in the same phrase is a bit redundant.

Response: Thanks for your comment. We have dropped 'more' in the expression as you suggested. We have rewritten the phrase 'In order to better exploit the modern development of geodetic techniques […]' to make it concise. The new phrase is 'In order to improve the accuracy and efficiency of the application in earth science based on space geodesy techniques, a reliable tropospheric delay model is required.'

3. line 40-57 – A (very) short description on the basic principles on which each model presented here is based (real time data, averaged trend from database, empyrical considerations...) would be helpful, along with their accuracy (which is already present).

Response: Thanks for your comment. We have shorten the description as you suggested, which is 'Some tropospheric delay models are developed to mitigate the tropospheric delay. The traditional models like the Hopfield model (Hopfield 1969), Saastamoinen model (Saastamoinen 1973) and Black model (Black 1978) require real-time meteorological data to reach a correction accuracy better than 10 cm. Given the location and time information, the UNB series models (Collins and Langley 1997, 1998; Leandro et al. 2006, 2008) and EGNOS model (Dodson et al. 1999; Penna et al. 2001; Ueno et al. 2001) use the empirical meteorological parameters in form of the latitude band table to estimate the ZTD with an accuracy of about 5 cm, while the IGGTrop model (Li et al. 2012) is based on the empirical three-dimensional parameters in form of the grids to calculate the ZTD with an accuracy of about 4 cm.'

4. ln 58 – '... required a number of' Which number? A large number of parameters? (If I correctly understand the context)

Response: Thanks for your comment. It is sorry to cause the confusion. The expression 'a number of' means 'numerous'. We have replaced it with 'a large number of' as you suggested.

5. ln 87 – 'the theoretical necessity'. I think that the use of analysis of data should show more the practical necessity, or the expression can just be omitted.

Response: Thanks for your comment. We have replaced 'the theoretical necessity' with 'the practical necessity'.

6. ln 116 – Do they represent the regions in the sense that they are representative? Or are they just example on how an expected diurnal cycle at different latitude should qualitatively be?

Response: Thanks for your comment. The six grid points are randomly selected at different latitude

regions to qualitatively show the global existence of the diurnal cycle.

7. The first time that day of the year (doy) is written, it should be written in the extended form.
Response: Thanks for your comment. We have written the extended form of doy.

8. Figure 2, captions and/or description – It is not very clear what error are represented by the vertical bars. Are they one standard deviation from the average?
Response: Thanks for your comment. The error bars denote the standard deviation from the average. We have added this explanation in the caption of Figure 2.

9. lines 171 - 179 – The fact that the new model is presented before its previous version makes this part not too easy to follow, at first. Maybe a swap between the two sentences, where the features of the older model are presented before the new ones could simplify things.
If I may add a comment, since the previous model has appeared only in (Yao et al.,2013) which is accessible only to chinese readers, a short introduction of this model at the beginning of this section (like the counterpart of formulae 1 and 2) can be useful.
Response: Thanks for your comment. We have added the short introduction of GZTD model at the counterpart of equation 1 and 2. The introductions are 'Our previous GZTD model only accounts for the annual and semi-annual variations of ZTD, whose first equation is similar to equation (1) but without the fourth term (diurnal term) on the right of equation (1).' and' The expansion equation of GZTD model is a 10-order and 10-degree spherical harmonic function which is 8 less order and degree than equation (2).'

10. line 181 – I don't get what this sentece mean, exactly. It is stated that the GZTD horizontal resolution is 18 degrees, but it is not clear if the diurnal variation above 5mm refers to the GZTD model or the expected diurnal variation which the model should usually solve. There are a couple of questions that should be answered, even if shortly, to clarify this part: how the horizontal resolution and ZTD resolution are related and what should be an optimal horizontal accuracy to get the most of the diurnal effects?
Response: Thanks for your comment. It is sorry to cause the confusion. The GZTD model doesn't have the diurnal term. The sentence means that the diurnal variations of real ZTD are very small while the 10 spherical harmonics adopted by GZTD model has a low resolution of about 18 ° which should be modified. We have rephrased this sentence as 'The 10 spherical harmonics adopted by GZTD result in a resolution of about 18 °, which are too low for GZTD2 model to reflect the small diurnal variations'. For the same temporal variation parameters to be expanded, the higher order and degree of spherical harmonics would get higher accuracy for expansion. The number of order and degree of spherical harmonics determine the horizontal resolution of model parameters associated with the ZTD estimates, which is the way how the horizontal resolution and ZTD resolution are related. The diurnal effects are quit small and inhomogeneous globally. The low resolution (>15 °) is hard to describe the features. However the higher resolution of model will bring more parameters. To keep the balance between accuracy and the number of parameters, we have chosen the moderate resolution (about 10 °) as the optimal one.

11. ln 214 – I would drop the last 'in theory'.

Response: Thanks for your comment. We have dropped the last 'in theory' in this line.

12. ln 265 – I am confused by the term 'climate change'. Do you mean that the statistics of the previous years that you include to set the parameter of GZTD2 and the other model are not representative of the year that you are using for the test? To me, the fact that the change in the ZTD has a relation with deep moist convection effects, whose specific occurence is impossible to predict using averaged data, and that are much more intense at the tropics, would seem sufficient to explain the increased error near the equator, at least qualitatively(so that they are related more to weather changes than to climate changes). If you are connecting it to climate change please explain better, possibly including some references.

Response: Thanks for your comment. We accepted your opinion that the increased errors near the equator are related more to weather changes than to climate changes. We have replaced 'climate change' with 'weather change' and rephrased the corresponding explanation as 'These areas are near the equator where the deep moist convection effects related to the change of ZTD are more intense , so the weather change in these areas are more complex compared with other areas, resulting in difficulty for modelling tropospheric delay.'

12. line 345 and line 368 – '... is obviously superior ...': I think that, even if 'obviously' is here clearly intended with the meaning of 'easily seen', its more common meaning of 'of course' can be a little misleading. I suggest to change it in some way.

Response: Thanks for your comment. For the clarity, we have replaced 'obviously' with 'substantially'.

13. ln 373 - 376 – This phrase needs some adjustment, with more clear logical connections between the part that are now simply separated by commas.

Response: Thanks for your comment. We have rephrased the sentence as 'In this paper, using the time series data of global tropospheric zenith delays provided by GGOS Atmosphere, we analyzed the diurnal variation in the ZTD which is neglected in the previous GZTD model, then we modified the model function to develop an improved model named GZTD2.'

14. ln 378 – The sentence starting with 'The testing results ...' is not very clear. I suggest to simplify and organize the informations better.

Response: Thanks for your comment. We have rephrased the sentence as' The testing results of GGOS ZTD grid data show that the global average Bias and RMS for GZTD2 model are 0.2 cm and 3.8 cm respectively. The global average Bias is comparable to that of GZTD model, but the global average RMS has been reduced by 0.3 cm. Both the Bias and RMS are far better than EGNOS model and the UNB series models.'

**Technical corrections**
Response: Thanks for your comment. We have accepted all your suggestions and made the corrections.
1. line 12 – GNSS → global navigation satellite systems (GNSS)
2. ln 20 – RMS → Root Mean Square (RMS)
3. ln 56 – Predication → Prediction

4. ln 72-73 – '... parameters for storage like *the* above grid models ...'

5. ln 79 – girds → grids

6. ln 146 – '..., and is featured ...': 'and' has been removed.

7. ln 162 – 'P nm is ... ' → 'P nm are ... '

8. ln 171 – 'Different from' → 'In contrast with' .

9. ln 215 – 'GZTD model ...' → 'The GZTD model ...'

10. ln 289 – 'followed' → 'follows' .

11. ln 325 – 'The reduce for RMS ...' → 'The reduction of RMS ...'

12. ln 340 – 'the Bias and RMS are lager with height less than 500 m ...' → 'the Bias and RMS are larger for height less than 500 m ...'

13. ln 369 – 'improve' → 'improves'

14. ln 377 – 'testing with ZTD grid data' → 'testing with GGOS ZTD grid data'

15. ln 383 – 'show the Bias ...' → 'show that the Bias'

16. ln 384 – 'that of GZTD' → 'those of GZTD'

---

## Author Comment (AC2) · 2 May 2016

point-by-point response to the comment

Scientific questions

1. Reading becomes tedious by overuse of acronyms and references to other models are results. Please, use cm or mm, not both.

Response: Thanks for your comment. We have reduced the overuse of acronyms and references to other models. Particularly, we have shorten the introduction about other models. The new second paragraph of the Introduction is 'Some tropospheric delay models are developed to mitigate the tropospheric delay. The traditional models like the Hopfield model (Hopfield 1969), Saastamoinen model (Saastamoinen 1973) and

Black model (Black 1978) require real-time meteorological data to reach a correction accuracy better than 10 cm. Given the location and time information, the UNB series models (Collins and Langley 1997, 1998; Leandro et al. 2006, 2008) and EGNOS model (Dodson et al. 1999; Penna et al. 2001; Ueno et al. 2001) use the empirical meteorological parameters in the form of the latitude band table to estimate the ZTD with an accuracy of about 5 cm, while the IGGTrop model (Li et al. 2012) is based on the empirical three-dimensional parameters in the form of the grids to calculate the ZTD with an accuracy of about 4 cm. However the IGGTrop model needs a large number of parameters. Then Li Wei et al. (2015) developed the new versions of IGGtrop named IGGtrop_ri (i = 1, 2, 3) by simplifying the algorithm and lowering the resolution, which substantially reduce the required numbers with a similar accuracy. Krueger (2004;2005) and Schüler (2014) obtained the annual and diurnal coefficients for underlying parameters by fitting every grid point's meteorological parameters time series of the National Centers for Environmental Prediction (NCEP) atmospheric data, and established two global tropospheric delay models-TropGrid and TropGrid2 . The correction accuracy of TropGrid2 is 3.8 cm. Böhm et al. (2015) proposed Global pressure and temperature 2 wet (GPT2w) as an extension to GPT2 (Lagler et al. 2013) with an improved capability to determine zenith wet delays in blind model. The GPT2w model accounts for the annual and semiannual variations of meteorological parameters, and the validation with IGS data and an extended validation with ray-traced delays (Möller et al. 2014) show a high accuracy of about 3.6 cm for GPT2w. However, GPT2w has numerous parameters for storage like the above grid models such as IGGTrop series models and TropGrid series models.'

In this paper, we have changed the unit of mm to cm in some parts.

2. I think that it's necessary an atmospheric approximation to problem. An approximation as of the article 'Seasonal variability of GPS-derived zenith tropospheric delay (1994–2006) and climate implications' of Shuanggen Jin, Jong-Uk Park, Jung-Ho Cho, Pil-Ho Park in Climate and Dynamics Journal can be more appropriate for this journal.

A reference to this article can be included.

Response: Thanks for your comment. The reference of article 'Seasonal variability of GPS-derived zenith tropospheric delay (1994–2006) and climate implications' is in the reference list of our paper. We have added a reference as you suggested, which is 'Pramualsakdikul, S., Haas, R., Elgered, G., & Scherneck, H. G. (2007). Sensing of diurnal and semi-diurnal variability in the water vapour content in the tropics using GPS measurements. Meteorological Applications, 14(4), 403-412.'

3. An article very similar has been written for Li Wei with their IGGTrop model for Chinese Science Bulletin, and comparisons with results of IGGtrop model (or TopGrid2) can be of interest when you compare your results. Please, change in your references and include the entire name of first author (line 447, line 449).

Response: Thanks for your comment. We have no access to the codes or programs of the IGGTrop model and TropGrid2 model. Maybe these models are not available for public currently. Although we cannot directly compare our model with these models, we presented these two models's accuracy verified in their articles in our introduction section. We have changed the reference and completed the entire name of first author as you suggested.

4. The aspects related with validation (3.2), can be the most contentious. Why you use these 362 IGS sites?

Response: Thanks for your comment. In 2010, some IGS sites have the severe problem of ZTD data missing. For a convinced validation, only the IGS sites with at least 120 days (approximately a third of the year) of tropospheric delays are selected. We have added this explanation in section 3.2.

5. Aspects related with the ZTD changes are poorly traited. Convection effects are much more intense at the tropics daily. This would seem sufficient to explain the increased error near the equator. In line 327 you compare between GZTD and GZTD2

and you indicate that there are improvements of RMS in areas where you have limited IGS sites to compare.

Response: Thanks for your comment. We have rewritten the explanation as 'These areas are near the equator where the deep moist convection effects related to the change of ZTD are more intense , so the weather change in these areas are more complex compared with other areas, resulting in difficulty for modelling tropospheric delay.' The expression of line 327 is not very rigorous as you pointed out, which has been rephrased as ' The significant improvements of RMS are found at the low-latitude sites which are distributed in Pacific Ocean, South America coast and West Africa coast where the diurnal variations are notable (see Figure 3d)'

Technical corrections

Change Predication in line 56 for Prediction. Change girds for grids in line 79. Include'2. The new.' in line 98 and '4. Conclusions' in line 372. You forgot the dot.

Response: Thanks for your comment. We accepted all your suggestions and made the corrections.

---

## Author Comment (AC3) · 10 May 2016

**The list of changes made in the manuscript**

1. line 12: add the full name of GNSS
2. line 21: add the full name of RMS
3. line 33: add a citation about iono-free combination techniques
4. line 37~40: rewrite the phrase
5. line 41~99: short the description and make some corrections
6. line 105: correct 'girds' to 'grids'
7. line 113: correct 'theoretically' to 'practical'
8. line 124: add the dot
9. line 144~145: add the extended form of doy
10. line 169~170: add the explanation for error bars
11. line 173: remove 'and'
12. line 193~196 and line 205~207: add the introduction of the GZTD model
13. line 200: replace 'Different from' with 'In contrast with'
14. line 212~216: rephrase the sentence to avoid confusion
15. line 247: remove 'in theory'
16. line 248: add 'The'
17. line 297~301: rewrite the sentence and add two citations
18. line 310~313: add the strategy of IGS sites selection
19. line 363: replace 'reduce' with 'reduction'
20. line 378: replace 'lager' with 'larger'
21. line 406: replace 'obviously' with 'substantially'
22. line 408: replace 'improve' with 'improves'
23. line 411: add the dot
24. line 412~431: rephrase the conclusions for clarity and make some corrections
25. line 493 and 495: add the full name of the first author
26. line 501~503 and 508~511: add three references

[revised manuscript text omitted]